# Enhancement of Skin Permeation and Penetration of β-Arbutin Fabricated in Chitosan Nanoparticles as the Delivery System

**Shariza Sahudin [1,\*], Nursyafiqah Sahrum Ayumi [1] and Norsavina Kaharudin [2]**

1   Department of Pharmaceutics, Faculty of Pharmacy, Universiti Teknologi Mara, Puncak Alam Campus, Shah Alam 42300, Selangor, Malaysia
2   VNI Scientific, Pusat Perniagaan Ixora, Senawang 70450, Negeri Sembilan, Malaysia
\*   Correspondence: shariza2280@uitm.edu.my; Tel.: +60-102210342

**Abstract:** Background: There has been an increase in demand for cosmetic skin-whitening products with efficacy toward lightening skin tone. β-arbutin is an inhibitor of tyrosinase enzyme activity within the skin's melanocytes, and so has shown considerable promise as a skin-lightening agent. It is, however, both hydrophilic and hygroscopic, which hinders its penetration of the skin to reach these melanocytes. Chitosan (CS) possesses considerable penetration-enhancing properties when utilized in topical delivery formulations. The strong affinity of positively charged chitosan nanoparticles toward negatively charged biological membranes can be exploited to achieve site-specific targeting. Objective: To investigate the use of chitosan nanoparticles (CSNPs) as carrier units to enhance the topical delivery of β-arbutin. Method: CSNPs containing β-arbutin were prepared using an ionic cross-linking method, and entrapment efficiency and loading capacity were evaluated at numerous β-arbutin concentrations. Further characterization involved using FTIR, XRD, TEM, and TGA, and in vitro permeation studies were conducted using in vitro Franz diffusion cells. Results: β-arbutin chitosan nanoparticles were successfully formulated with a size range of 211–289 d.nm, a polydispersity index between 0.2–0.3, and zeta potential in the range 46.9–64.0 mV. The optimum encapsulation efficiency (EE) and loading capacity (LC) of β-arbutin were 68% and 73%, respectively. TEM revealed the nanoparticles to be spherical in shape. FTIR spectra revealed characteristic chitosan-related peaks appearing at 3438.3 $cm^{-1}$ (-OH stretching) and 3320 $cm^{-1}$ (-CH stretching), together with 1598.01 $cm^{-1}$ (-NH$_2$) specific to β-arbutin nanoparticles. XRD analysis revealed an increase in crystallinity and TGA analyses identified increasing thermal stability with increasing β-arbutin concentration. In vitro studies indicated higher permeation and improved penetration of β-arbutin loaded in CSNPs compared to its free form. Conclusion: CSNPs present considerable promise as effective carriers for improved topical delivery of β-arbutin.

**Keywords:** chitosan; chitosan nanoparticles; arbutin; beta-arbutin; percutaneous delivery; whitening agent; cosmetic active





## 1. Introduction

Arbutin (β-arbutin) is a skin-lightening and depigmentation agent found in various over-the-counter products, such as creams and lotions available in Japan and the United States. It is obtained naturally from the dried leaves of various berry species such as bearberry, cranberry, and blueberry [1,2]. Arbutin, a b-D-glucopyranoside derivative of hydroquinone, has been found to inhibit the biosynthesis of melanin through the inhibition of tyrosinase activity. This inhibition of tyrosinase activity is dose-dependent and occurs at noncytotoxic concentrations [3]. It also inhibits melanosome maturation and is less cytotoxic to melanocytes than hydroquinone. β-arbutin also possesses antioxidant, anti-inflammatory, and antitumor properties [4]. The Scientific Committee on Consumer Safety (SCCS) recommends limits to the amount of β-arbutin used in cosmetics, 2% in face creams and 0.5% in body lotions [5,6]. The demand for cosmetic skin-whitening products is

particularly high among populations in South and Southeast Asian countries [7,8]. Arbutin is hydrophilic and hygroscopic, with a log *p* value of −1.49. These present a barrier to the absorption of arbutin into the stratum corneum in order to reach the melanocytes. Thus, the use of chitosan-derived nanotechnology [7] attempts to improve the penetration and permeation of arbutin into the skin.

Chitosan (CS) is a chitin-derived alkaline deacetylated bipolymer of N-acetyl glucosamine, consisting of β-(1, 4)-linked-D-glucosamine residues with randomly acetylated amine groups [9]. Due to its biocompatibility and biodegradability, CS has been investigated for a number of potential pharmaceutical and medical applications [3,9–11]. CS has also received attention due to its mucoadhesive [9] and controlled-release properties [12]. However, its permeation-enhancing properties [13,14] have resulted in extensive research for potential use in topical delivery formulations [15,16].

In order to maximize its effectiveness within drug delivery systems, CS must be both hydrosoluble and positively charged [17]. This facilitates the interaction of CS within an aqueous environment with negatively charged polymers, macromolecules, polyanions, biological membranes, and of course the skin, which is negatively charged at neutral pH [17,18]. Compared with negatively charged species, positively charged nanoparticles have been found to be far more effective in improving a drug's penetration through the skin [19]. For cosmetic formulations, sustained-release properties are another useful therapeutic option that facilitates prolonged dermal therapy [20].

This present study attempts to formulate CS nanoparticles with optimum particle size, zeta potential, polydispersity index (PdI), high encapsulation efficiency (EE), and loading capacity for use as potential carrier units for the topical delivery of β-arbutin. Morphological properties of the nanoparticles were evaluated using TEM and physical characteristics were examined using Fourier transform infrared (FTIR), X-ray diffraction (XRD), differential scanning calorimetry (DSC), and thermogravimetric (TGA) analyses. Finally, in vitro permeation and penetration studies utilizing rat skin were conducted to evaluate the efficiency of this nanoparticle system in delivering β-arbutin through the skin.

## 2. Materials and Methods

All chemicals and reagents used were of an analytical grade.

### 2.1. Materials

95% deacetylated CS with low molecular weight of 50,000–190,000 D, β-arbutin, and sodium triphosphate (TPP) were sourced from Sigma Aldrich (St. Louis, MO, USA). Phosphate buffer solution (PBS) of pH 7.4 and HPLC-grade acetonitrile were purchased from Fisher Scientific, Loughborough, UK.

### 2.2. Preparation of β-Arbutin Loaded CSNPs

β-arbutin CSNPs were prepared by ionic crosslinking [21,22] with CS using TPP. A 2 mg/mL solution of CS was prepared using glacial acetic acid. A 1 mg/mL solution of TPP was also prepared. β-arbutin was added in varying concentrations (0.1, 0.2, 0.4, and 0.5%) to 12 mL of TPP to produce a β-arbutin/TPP primer solution. Nanoparticles were spontaneously formed by the dropwise addition of this primer to the CS solution.

A total of 12 mL of β-arbutin/TPP primer solution was mixed with 25 mL of CS solution and magnetically stirred (600 rpm) at room temperature for 2 h. The resulting β-arbutin CSNPs were separated by ultracentrifugation at 30,000 rpm, 25 °C for 45 min, and then lyophilized (Scanvac CoolSafe 110, Chemoscience, Bangkok, Thailand) at −90 °C for 24 h.

### 2.3. Evaluation of Polydispersity Index, Particle Size, and Zeta Potential

The resulting nanoparticles were re-dispersed in distilled water [23] before being analyzed at 25 °C using a Malvern Zetasizer 1600 Nano ZS at a detection angle of 90° (n= 3).

*2.4. Evaluation of Loading Capacity (LC) and Entrapment Efficiency (EE)*

A standard calibration curve for β-arbutin was produced using reverse-phase HPLC at the determined optimum absorption wavelength of 286 nm.

Entrapment efficiency and loading capacity percentages were determined as follows:

$$EE\ (\%) = (\frac{Wt - Wf}{Wt}) \times 100$$

$$LC\ (\%) = (\frac{Wt - Wf}{Wn}) \times 100$$

*Wt* and *Wf* represent total and free β-arbutin respectively, and *Wn* is the total weight of lyophilized β-arbutin CSNPs.

*2.5. TEM Analysis*

A morphological examination of lyophilized β-arbutin CSNPs was conducted using transmission electron microscopy (TEM). Nanoparticles were dispersed in acetone and sonicated for 15 min. One drop of each sample was placed on a copper microgrid, stained by phosphotungstic acid, air-dried at room temperature (25 ± 2 °C), coated with gold, and then observed under the electron microscope at 10,000 to 100,000 magnification.

*2.6. FTIR Analysis*

The FTIR spectra for β-arbutin CSNPs were obtained using an FTIR spectrophotometer (Spectrum 100; Perkin Elmer, Waltham, MA, USA). An amount of 2 mg to 3 mg of β-arbutin CSNPs, CS, or β-arbutin was mixed with 80 mg to 90 mg of potassium bromide (KBr) and compressed using a hydraulic press to form a transparent pellet. The resulting pellets were scanned in the spectral region of 400–4000 cm$^{-1}$ at a resolution of 4 cm$^{-1}$ under transmission mode.

*2.7. XRD Analysis*

The degree of crystallinity of β-arbutin CSNPs was evaluated using a BEDE D-3 Cu Kα X-ray crystallographic diffractometer (Bruker, D-8 advance, Billerica, MA, USA), utilizing a monochromatic voltage of 40 kV and a generator current of 100 mA. The samples were scanned at a rate of 10 °C/min from 2θ = 1–100°.

*2.8. TGA Analysis*

Thermogravimetric analysis (Perkin Elmer Model, Waltham, MA, USA) was conducted by utilizing a 2 mg to 3 mg mass sample of β-arbutin CSNPs placed in a balance-based sample pan. The mass of samples was evaluated as a function of temperature (25 °C to 800 °C) using a nitrogen flow rate of 50 cm$^3$/min

*2.9. In Vitro Permeation Study*

Permeation studies were performed with full-thickness Wistar rat skin using a method described by Singka et al. (2010) [20]. Freshly obtained rat skin was acquired from the UiTM LAFAM animal unit. The skin was shaved and then excised and cut into 2 cm × 2 cm sections. Subcutaneous fat was removed to ensure smooth permeation and then the skin was cleaned under running water. The skin was immersed in PBS prior to mounting on Franz diffusion cells (Permegear, Cranford, NJ, USA). The skin sections were mounted on diffusion cells with the dermal side facing a receptor compartment filled with PBS. A specific dosage of β-arbutin CSNP cream was fully placed into the donor chamber with the help of a glass rod and stirred at a constant rate. Sampling was carried out for 24 h at specific time intervals and the receptor solution was analyzed using HLPC. A standard calibration curve was constructed by preparing a stock solution of β-arbutin in PBS (1–0.004 mg/mL).

### 2.10. In Vitro Penetration Study

The tape stripping method was utilized to characterize penetration by determining the amount of β-arbutin remaining on the skin's surface after application. The skin's surface was stretched using 1.5 cm × 2 cm adhesive tape and a controlled weight. Each stripped tape was then suspended in 4 mL of ethanol (96% *v/v*) in an individual glass vial to extract β-arbutin from the skin. This was repeated to obtain 20 individual tapes [21]. The epidermis was separated from the dermis at 50 °C using a hot plate. Separate glass vials were used to place the remaining epidermis. Sections of the dermis were suspended in ethanol after being cut into small pieces and before undergoing homogenization. The glass vials were then placed in a shaking water bath at 37 °C overnight prior to HPLC analysis. A stock solution was prepared using ethanol to construct a standard calibration curve.

### 2.11. RP-HPLC System

The chromatographic system consisted of RP-HPLC (Waters 600 controller, in-line degasser AF, Autosampler 2707, and Photodiode Array Detector 2998, USA) and a Waters Symmetry C18 column (250 × 4.5 mm; 5 m). A mobile phase of acetonitrile and water at a ratio of 20:80 (*v/v*) delivered at a flow rate of 1 mL/min with an injection volume of 10 μL. Calibration curves were constructed within a β-arbutin concentration range of 0.1–1.3 g/mL, analyzed at a maximum wavelength of 272 nm, and retention time of 2.2 min. Standard curves were obtained by plotting the peak areas of the relevant concentration of β-arbutin and subjected to regression analysis, resulting in a correlation determination (R2) of 0.9916.

### 2.12. Statistical Analysis

Data were analyzed using the Statistical Package for the Social Sciences (SPSS, Version 26.0 SPSS Inc., Chicago, IL, USA). All data presented with calculated mean, $\pm$ standard deviation, and *p*-values using either paired *t*-tests or one-way ANOVA, followed by Tukey's post-hoc analysis, where a *p*-value < 0.05 indicated statistical significance.

### 3. Results and Discussion

### 3.1. Physicochemical Characterization β-Arbutin CSNPs

The particle size distribution of β-arbutin CSNPs are shown in Figure 1 and are in the range of 211 nm to 289 nm. The particle sizes of β-arbutin CSNPs (0.1–0.6%) increase as the concentration of β-arbutin increases. This phenomenon was also reported by various researchers [22–24]. Thus, as the β-arbutin concentration and CS ratios were increased from 0.1–0.5%, the size of the nanoparticles also increased. The particle sizes of CSNPs produced using 0.6% β-arbutin were significantly larger than those from 0.1% and 0.2% (*p* < 0.05 both). A previous study by Lademann et al., (2011) revealed that particles nanometers (nm) in size were shown to penetrate deeply into the skin, where they remained for up to 10 days within the stratum corneum [10]. Furthermore, Falguni Pati et al., (2011) reported that particles with smaller diameters penetrated better into human skin, evaluated using biopsies and analyzing the histological sections of the skin [3]. A comparison between particulate-sized and non-particulate-sized formulations revealed better penetration into the stratum corneum in the former.

In an acidic environment, chitosan (which has a pKa of 6.5) is polycationic. The addition of negatively charged crosslinking agents such as the TPP to the chitosan-drug mixture leads to the formation of chitosan nanoparticles. Crosslinking agents such as TPP contain both -OH and phosphoric species, which undergo electrostatic and hydrogen bond interactions with $NH^{3+}$ sites on chitosan. Thus, the formation of the nanoparticles is a function of the number of free amino groups present, which serves to facilitate the formation of electrostatic interactions between nanoparticles and drugs. The effectiveness of such interactions also helps to reduce nanoparticle size, which is important since as drug loading increases, particle size also increases [25–27]. The increase in nanoparticle size could also be due to the incorporation of cations within the nanoparticles. By increasing

the amount of CH, the dispersion viscosity also increases, however, the exerted shear is insufficient to result in a reduction in particle size [28–30]. Figure 2 shows the polydispersity index of all nanoparticles. No significance difference was seen among all subsets.

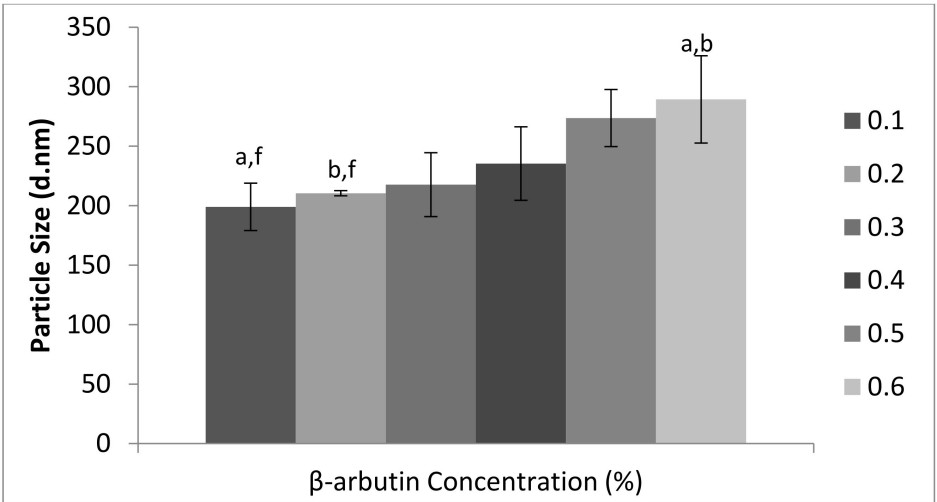

**Figure 1.** Particle size of 0.1%, 0.2%, 0.3%, 0.4%, 0.5%, and 0.6% β-arbutin CSNPs. (*n* = 3). Different alphabets show significance difference, *p* < 0.05.

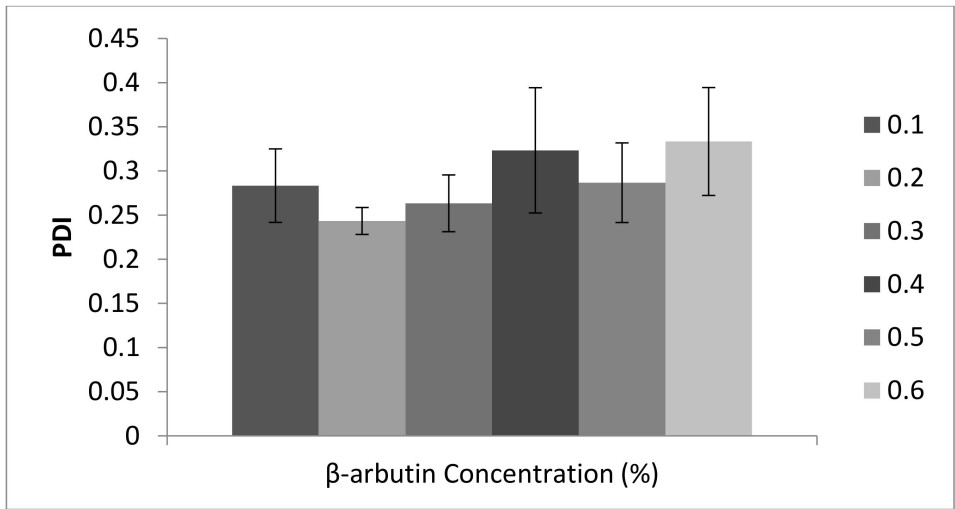

**Figure 2.** Polydispersity index of 0.1%, 0.2%,0.3%,0.4%, 0.5%, and 0.6% β-arbutin CSNPs. (*n* = 3). No significance difference among all subsets.

The zeta potential is an important parameter used to characterize the stability of dispersion systems [26]. The zeta potential describes the electrokinetic potential of particles within colloidal systems and is defined as the potential difference between the shear location, corresponding to the surface of a tightly bound surface layer, and bulk solution [27]. As the zeta potential approaches 0 mV, inter-particulate repulsion reduces, and so a dispersion becomes less stable since particles are better able to approach one another to interact and form aggregates, a sign of lower dispersion stability [4,14]. A flocculated dispersion, consisting of loose aggregates, will typically have a zeta potential within the range of −20 to +20 mV. Thus, increasing the zeta potential beyond this will improve overall dispersion stability against aggregation [27]. In summary, greater repulsive forces between particles prevents aggregation and improves dispersion stability [31,32]. The zeta potentials (Figure 3) of the β-arbutin CSNPs remain within acceptable limits, which contribute to overall stability.

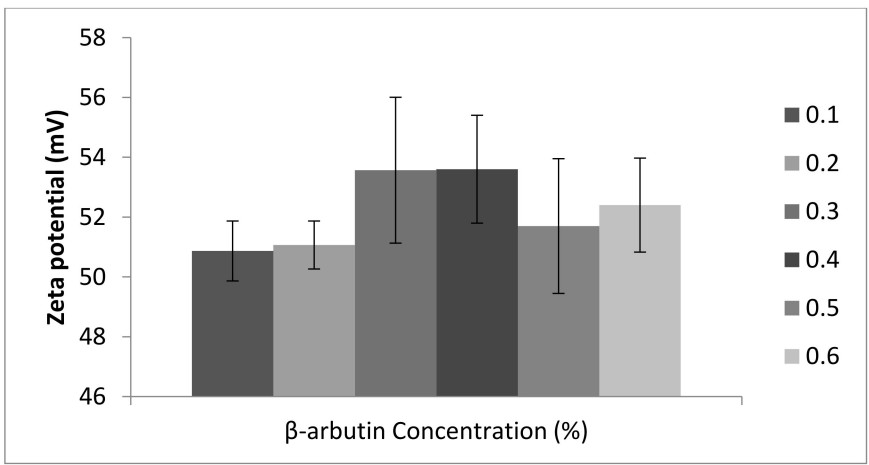

**Figure 3.** Zeta potential of 0.1%, 0.2%,0.3%,0.4%, 0.5%, and 0.6% β-arbutin CSNPs. ($n = 3$). No significance difference among all subsets.

The cellular uptake of nanoparticles is a two-stage process first involving a binding step onto the cell membrane, followed by an internalization step [25]. Both steps are influenced by particulate properties such as size, shape, and surface charge. Cellular surfaces are dominated by negatively charged molecules, and so the ability of nanoparticles to interact with cell membranes is significantly influenced by surface charge [26–28]. This ultimately facilitates cellular uptake and directs NPs to specific cellular compartments both in vitro and in vivo [24,25]. The cationic charge of chitosan-based nanoparticles favors their electrostatic interaction with negatively charged tissues and cell membranes. After the electrostatic interaction-mediated adsorption of the nanoparticles onto a cellular membrane, uptake can occur via several mechanisms including pinocytosis, receptor-mediated endocytosis, or phagocytosis [31]. Particle size influences the ability of nanoparticles to uptake into and across a membrane, and hence impact their ability to penetrate into the skin. Thus, both surface charge and size influence the ability of nanoparticles to penetrate into the skin [32–34].

### 3.2. Efficiency (EE) and Loading Capacity (LC) of β-Arbutin CSNPs

The EE (%) and LC (%) of β-arbutin was determined to be 68% and 73.4%, respectively, for a 0.4% β-arbutin concentration (Table 1).

**Table 1.** Entrapment efficiency and drug loading capacity of β-arbutin CSNPs.

| β-Arbutin Concentration | Entrapment Efficiency (%) | Loading Capacity (%) |
|---|---|---|
| 0.1% | 43.3 | 56.2 |
| 0.2% | 45.0 | 45.2 |
| 0.3% | 50.3 | 47.0 |
| 0.4% | 68.0 | 73.8 |
| 0.5% | 54.3 | 58.1 |
| 0.6% | 56.3 | 61.9 |

Entrapment efficiency is influenced by fabrication processes, preparation, methods, the physiochemical properties of β-arbutin [13,35–37], and numerous formulation variables [13]. In the present study, the use of β-arbutin concentrations of 0.5% and 0.6% caused a slight reduction in the drug-loading capacity of the nanoparticles. It appears both encapsulation and loading capacities reach an optimum at 0.4% β-arbutin concentration. Increasing the formulation ratio between β-arbutin and CS leads to an improvement in entrapment efficiency due to the formation of larger particles with reduced surface area, resulting in a reduced diffusion of the drug out of the system [38–41]. Increasing CS concentration reduces the encapsulation efficiency of β-arbutin [42,43] and increasing

CS concentration to 4 mg/mL hinders encapsulation [44,45] due to the higher solution viscosities resulting from higher CS concentrations.

### 3.3. Morphology

TEM of β-arbutin CSNPs was utilized to assess morphology and uniformity. The resulting TEM monograph of β-arbutin CSNPs (Figure 4) indicates spherically shaped nanoparticles in a near mono-dispersed system with a particle size range of 39.3 to 318 nm. Zetasizer analysis also demonstrated an acceptable range size of 120 nm to 300 nm. Previous published studies report similar spherical, rough-surfaced nanoparticles [20,46,47]. In addition, aggregation behavior is observed in some nanoparticles, and this could be due to interactions at substrate surfaces occurring upon drying or the result of volatilization of solvent on substrate surfaces, leading to interactions [48,49].

### 3.4. FTIR

FTIR analysis was performed on β-arbutin CSNPs within the concentration range 0.1–0.6% in order to isolate the bonding within the chitosan and β-arbutin nanoparticle matrix. The FTIR spectra reveal a strong, wide O-H bond stretching vibration peak at 3500–3000 cm$^{-1}$ and 3750–3000 cm$^{-1}$. These overlap with stretching vibrations between N-H with -CH$_2$ and -CH$_3$ moieties, which occur at 3320 cm$^{-1}$ and 3438 cm$^{-1}$, respectively.

During the development of β-arbutin CSNPs, the complexation of β-arbutin/TPP with chitosan occurs through the formation of ionic bonds, in which positively charged amine groups within the chitosan interact with negatively charged carboxylate groups in the β-arbutin/TPP primer system. This electrostatic interaction results in a spectral shift of the amine group from 1150 cm$^{-1}$ to 1170 cm$^{-1}$. Within the resulting β-arbutin CSNPs, the spectra reveal a widening and shift of the -OH peak from 3438 cm$^{-1}$ to 3320 cm$^{-1}$. In addition, an increase in the relative intensity of this peak suggests enhancement of hydrogen bonding between the two reactive species. The N-H bending vibration peaks for amine-I at 1600 cm$^{-1}$ and the amide-II carbonyl stretch at 1650 cm$^{-1}$ shifts to 1540 cm$^{-1}$ and 1630 cm$^{-1}$, respectively. The cross-linked chitosan-TPP complex also reveals a P=O peak at 1170 cm$^{-1}$. This suggests a strongly linked interaction between the TPP phosphoric and chitosan ammonium ions of the two reactive systems [34].

### 3.5. XRD Analysis

XRD diffractograms of the various β-arbutin CSNP concentrations are shown in Figure 5. The XRD pattern of samples showed peaks in the range of 2θ–30θ. The XRD diffractograms of free β-arbutin indicates a more amorphous structure in comparison to blank CSNPs, in which a prominent crystalline crystal structure is observed. The addition of β-arbutin into CSNPs results in a mixture consisting of both an amorphous form and crystals. This structural modification could potentially be due to an intermolecular and/or intramolecular network structure of CS, crosslinked to each other by TPP counterions and free β arbutin. These interpenetrating polymer chains imply chain alignment and consequent increase in crystallinity of β-arbutin CSNPs compared to free β-arbutin.

### 3.6. TGA Analysis

Thermograms of chitosan nanoparticles revealed decomposition patterns emerging at approximately 150 °C and terminating at approximately 330 °C (Figure 6). The data indicate a degree of thermal stability of β-arbutin CSNPs that can be directly related to changes in the structure indicated above from amorphous form for free CSNPs to an amorphous/crystalline mixture for β-arbutin CSNPs. It was also observed that β-arbutin CSNPs with β-arbutin at concentrations of 0.1%, 0.4%, and 0.6% possess better thermal stability compared to the other concentrations. This can be seen from weight loss curves at 250–300 °C, while for the other concentrations, the weight loss curves occur at 150–200 °C.

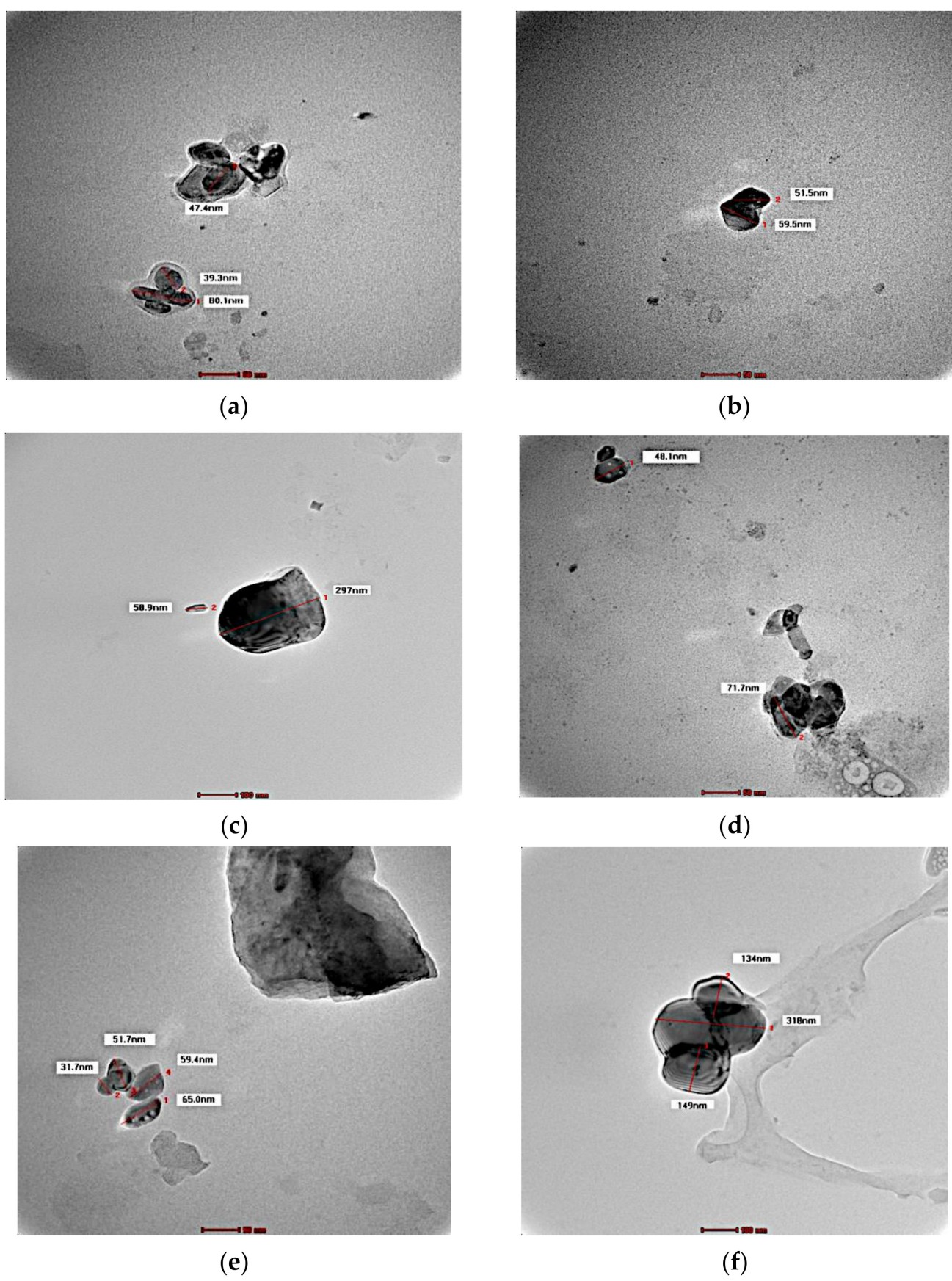

**Figure 4.** Transmission electron microscopy at 86,000× of 0.1% (**a**), 0.2% (**b**), 0.3% (**c**), 0.4% (**d**), 0.5% (**e**), and 0.6% (**f**) β-arbutin CSNPs.

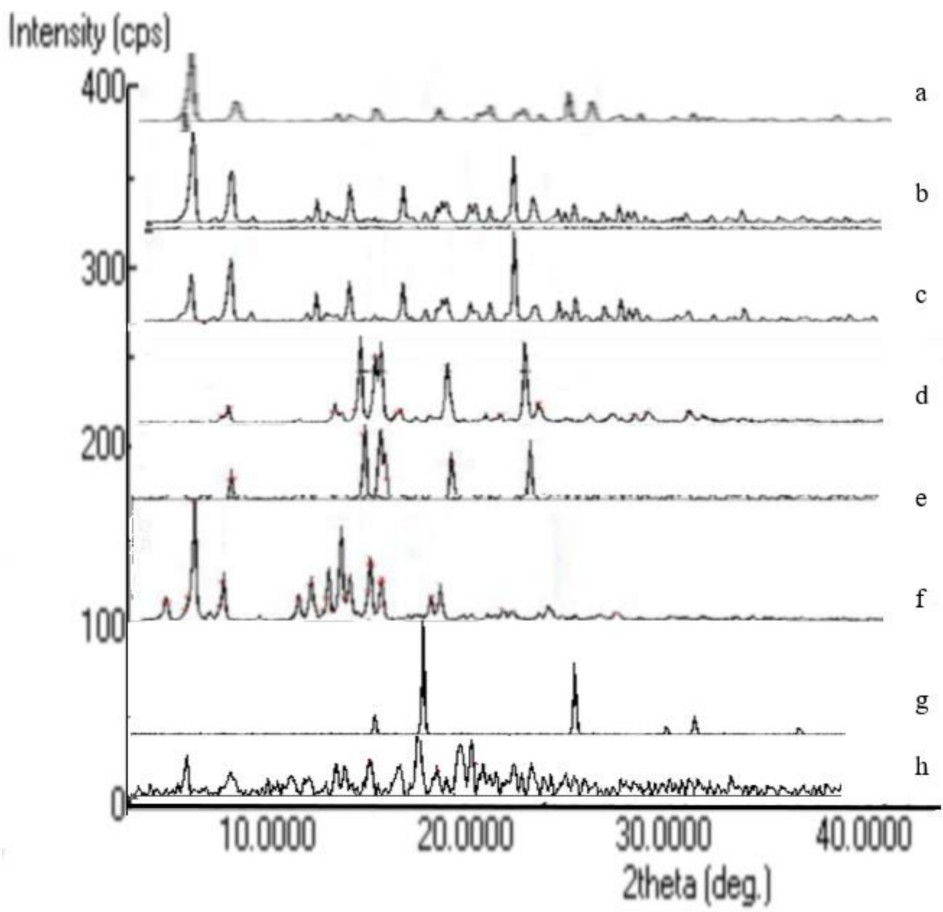

**Figure 5.** XRD analysis of 0.1% β-arbutin (a), 0.2% β-arbutin (b), 0.3% β-arbutin (c), 0.4% β-arbutin (d), 0.5% β-arbutin (e), 0.6% β-arbutin (f), blank CSNPs (g), and free β-arbutin (h).

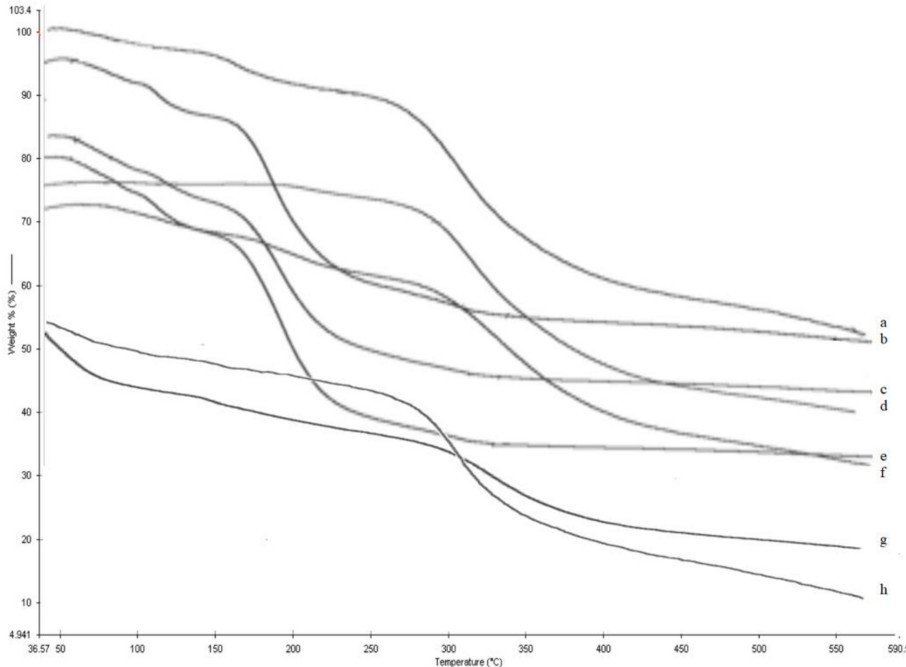

**Figure 6.** Thermograms of 0.1% β-arbutin (a), 0.2% β-arbutin (b), 0.3% β-arbutin (c), 0.4% β-arbutin (d), 0.5% β-arbutin (e), 0.6% β-arbutin (f), CSNPs, free β-arbutin (g), and blank CSNPs (h).

### 3.7. In Vitro Permeation Study

Permeation studies in rat skin were conducted over 72 h using 0.4% β-arbutin CSNPs, previously determined to possess the highest EE and LC. The results for the percentage of drug permeation for both β-arbutin CSNPs and free β-arbutin are shown in Figure 7. Based on the obtained results, the percentage of permeation of β-arbutin CSNPs was significantly higher than its free form. In the first 6 h, the percentage of permeation of β-arbutin CSNPs was around 6.4%, which slowly increased with time. The percentage permeation of free β-arbutin was low, 0.6% for the first 6 h. After 24 h, the percentage permeation of β-arbutin CSNPs was 11.1%, in comparison to the free form which was only 0.9%. After 72 h, β-arbutin CSNPs' percentage permeation through the skin reached approximately 34% and was significantly higher than the permeation of free β-arbutin (where $p < 0.05$). β-arbutin permeation comparisons were also performed at 12, 24, and 48 h, and the results indicated a significantly greater permeation of β-arbutin from CSNPs than its free form.

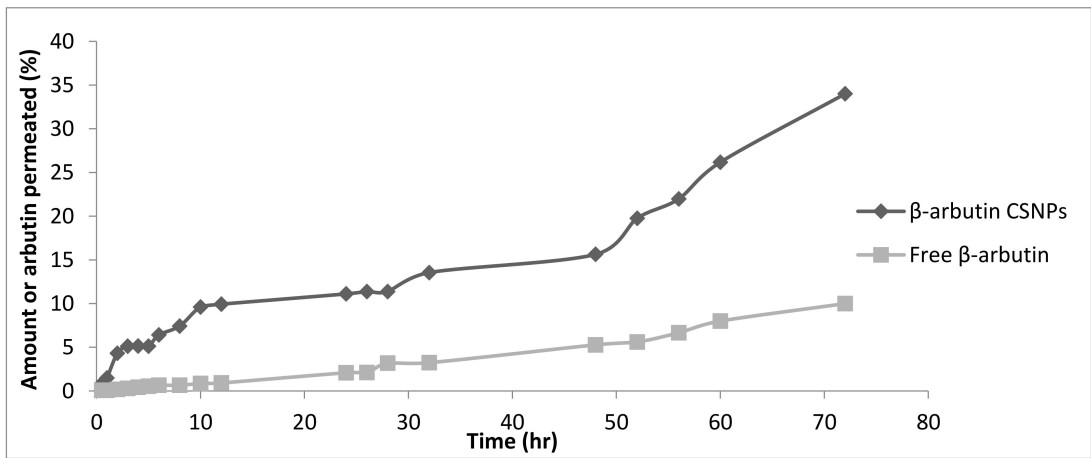

**Figure 7.** In vitro permeation studies of 0.4% β-arbutin CSNPs and free β-arbutin. (*t*-test, $p < 0.05$).

Chitosan has been evaluated for a number of pharmaceutical applications, including within dermal and transdermal delivery systems [1]. Numerous studies have reported enhanced skin delivery of numerous drugs following the use of chitosan as nanoparticle carriers. A number of mechanisms have been proposed by which chitosan increases the delivery of drugs across the skin. These include the interaction of vesicles with the skin's surface and its components, facilitated entry into the intercellular lipid matrix of the stratum corneum (SC), modification of lipid lamellae [2], and an influence on trans-epidermal osmotic pressure [4].

The skin acts as a negatively charged membrane [6,22] due to the presence of a negatively charged lipid layer. The presence of these charges within the skin membrane influences the transdermal deposition of drugs. Positively charged species such as chitosan exhibit higher permeability compared with negatively charged molecules. Recent formulation approaches have confirmed this penetration enhancement compared with rigid structure vesicles [33].

The skin's appendages also serve as potential points of entry and penetration into the SC [18]. Funnel-shaped hair follicles, for instance, extend up to 200 nm into the skin [35]. Their large surface area disrupts the epidermal barrier within the lower regions of the skin structure and act as efficient reservoirs for drugs and nanoparticles. Any species located within these structures can continuously diffuse to the surrounding spaces, cross the capillary walls, and potentially reach the circulation system [24]. Bostanudin et al., reported their amphiphilic-modified guar gum nanoparticles enhanced cell membrane penetration of arbutin, resulting in improved absorption, which is consistent with the literature [50]. It has been reported that nanoparticles with a diameter in the range between 0.25–3 mm were found to exhibit optimum phagocytosis, while lower size ranges are either

absorbed via caveolin- or clathrin-dependent endocytosis [51]. The presence of mannose receptors on human keratinocytes was also found to aid in the cellular internalization of carbohydrate-based nanoparticles [52,53].

### 3.8. In Vitro Penetration Study

A standardized skin stripping technique was utilized to assess the ability of CSNPs to deliver β-arbutin into various layers of the stratum corneum. Permeation results reveal the amount of β-arbutin within the stratum corneum layers (Figure 8), and it can be seen that the release rate of free arbutin at 72 h remains constant. No overshoot of release is demonstrated if or when the integrity of the skin is compromised. Based on the results, the concentration of β-arbutin from CSNPs present within the skin layers was found to be less in comparison to β-arbutin in its free form. Figure 8a clearly shows that larger quantities of free β-arbutin were found in the layers of the stratum corneum. The total amount of β-arbutin in the stratum corneum was approximately 17.1 mg for β-arbutin CSNPs (63%) compared with 22.4 mg (83%) of free β-arbutin, calculated from the total loaded β-arbutin, which was 27 mg. The results complemented our permeation studies, in which β-arbutin in CSNPs demonstrated greater permeation through the skin than the free form.

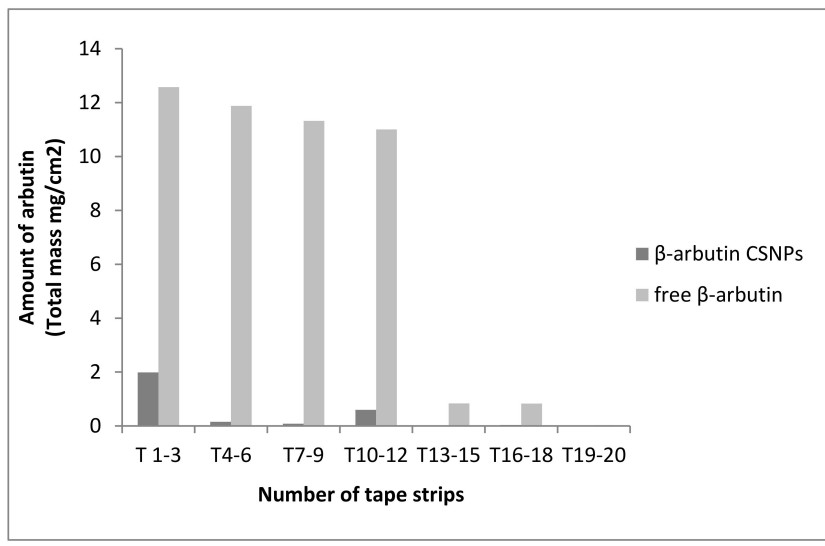

(a)

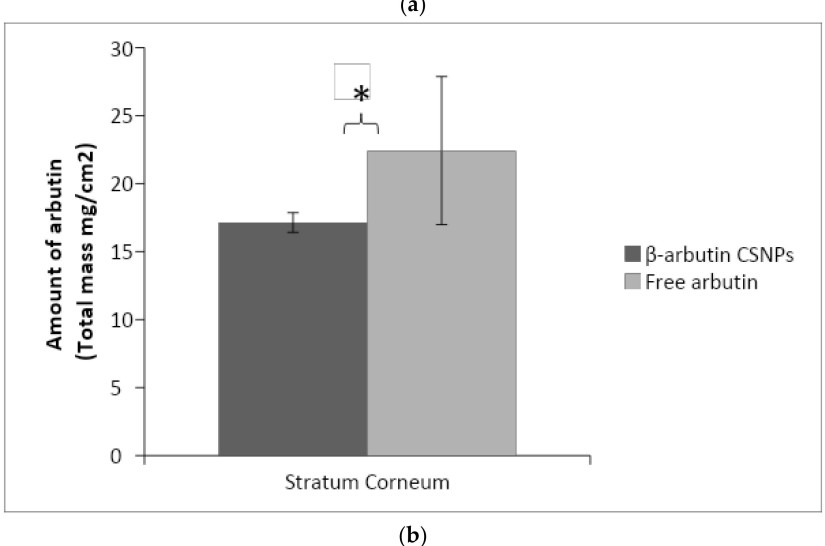

(b)

**Figure 8.** In vitro penetration study of 0.4% β-arbutin CSNPs and free β-arbutin. (**a**) Amount of arbutin found in the different tape strips and (**b**) Total amount of arbutin in the stratum corneum. (* indicates a significance difference, $p < 0.05$).

Possible mechanisms for the improved penetration and permeation of β-arbutin in nanoparticles include its ability to promote skin penetration through an interaction with the stratum corneum, leading to possible modification of the intercellular lipid lamellae. This facilitates partitioning of the CSNPs through the SC [37,38]. Negatively charged components within the skin lead to interaction with positively charged chitosan within the CSNPs, leading to enhanced drug penetration [29,39]. Song and Kim et al., reported higher drug penetration from positively charged chitosan-derived systems, compared to neutral and negatively charged chitosan [40].

## 4. Conclusions

β-arbutin CSNPs were successfully developed using a CS-TPP gelation method in which 0.1–0.6% of β-arbutin was comparatively loaded. TEM, XRD, FTIR, and TGA suggest that 0.4% of β-arbutin gave the best characteristics. At 0.4% concentration, β-arbutin-loaded CSNPs with an average particle size of 220 nm, PDI of 0.33, and zeta potential of 54 mV with EE of 68% and LC of 73.8% were obtained, with about five times higher permeation with significantly lower β-arbutin retention on the stratum corneum compared to the unbound or unloaded form. Chitosan nanoparticles have been shown to be suitable for use as a carrier-based delivery system for the topical application of β-arbutin. Further studies are required to ascertain the side effects of using these nanoparticles in vivo using animal studies.

**Author Contributions:** Conceptualization, S.S.; methodology, S.S.; data curation, N.S.A.; writing—original draft preparation, N.S.A.; writing—review and editing, S.S.; supervision, S.S. All authors have read and agreed to the published version of the manuscript. N.K.; Part Funding of studies and publication.

**Funding:** PYPA Grant from RMC UiTM/UCS UiTM and VNI Scientific (100-TNCPI/PRI 16/6/2 (056/2022)).

**Institutional Review Board Statement:** Not applicable.

**Informed Consent Statement:** Not applicable.

**Data Availability Statement:** All the data are available in the manuscript.

**Acknowledgments:** The authors would like to acknowledge the Faculty of Pharmacy, UiTM for providing scientific equipment and facilities for this work. PYPA Grant from RMC UiTM/UCS UiTM and VNI Scientific (100-TNCPI/PRI 16/6/2(056/2022)) for funding.

**Conflicts of Interest:** The authors declare no conflict of interest.

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
