# Peer review of "Enhancement of Skin Permeation and Penetration of β-Arbutin Fabricated in Chitosan Nanoparticles as the Delivery System"

_cosmetics, doi:10.3390/cosmetics9060114_

Round 1
Reviewer 1 Report
The study is clear and welldone treating a topic of Great interest for dermatologists and Cosmetic chemists. It will be necessary further studies to verify the eventual topic side effects due to to the arbutin.
Author Response
Yes, I agree with the reviewer. I have added in the conclusion for further studies to ascertain the side effects of the nanoparticles in vivo using animal studies

Reviewer 2 Report
Comment 1
Please specify the category of products (e.i. drugs or cosmetics) for the data given in the first sentence of the Introduction.
Comment 2
For In vitro Permeation study provide the information of the skin origin.
Comment 3
Which statistical program was used for analysis?
Author Response
Comment 1: I have specified creams and lotions
Comment 2: I have specified skin of wistar rats
Comment 3: I have specified using the Statistical Package for the Social Sciences (SPSS, Version 26.0 SPSS Inc., Chicago, IL, USA) for statistical analysis.
